# Preventing Agricultural Non-Point Source Pollution in China: The Effect of Environmental Regulation with Digitization

**DOI:** 10.3390/ijerph20054396

**Published:** 2023-03-01

**Authors:** Weikun Zhang, Peng Gao, Zhe Chen, Hailan Qiu

**Affiliations:** 1School of Social and Public Administration, Lingnan Normal University, Zhanjiang 524088, China; 2School of Management, Guangdong Ocean University, Zhanjiang 524088, China; 3School of Economics & Finance, Zhanjiang University of Science and Technology, Zhanjiang 524088, China; 4School of Economics and Management, Jiangxi Agricultural University, Nanchang 330044, China

**Keywords:** agricultural non-point source pollution, environmental regulation, digitization, geographic detector tool

## Abstract

Environmental regulation (ER) is essential to preventing agricultural non-point source pollution (ANSP). Prior research has focused on the effect of ER on agricultural pollution (AP), but little is known about the impact of ER following digitization on preventing AP, particularly ANSP. Based on the spatial heterogeneity, the effect of ER was examined using a geographic detector tool with provincial panel data from 2010 to 2020 in rural China. The results show that ER is a driver in preventing ANSP, primarily because of the constraint on farmers’ behavior. Digitization positively affects the prevention of ANSP, as the new impetus for the infrastructure, technology, and capital is supported. The interaction between ER and digitalization forms a driving effect on the prevention of ANSP, indicating that digitalization constitutes the path dependence of farmers’ rule acquisition and perception and addresses the “free riding” dilemma of farmers’ participation, thereby enabling the incentive of ER to make agricultural production green and efficient. These findings indicate that the endogenous factor of digitization allowing ER is essential to preventing ANSP.

## 1. Introduction

Agricultural non-point source pollution (ANSP) refers to the excessive accumulation of organic matter in soil or water bodies due to rainfall and topography, resulting in excessive chemical inputs in planting and crop straws, and the improper disposal of livestock and poultry manure in aquaculture, causing damage to the ecological environment. Pollution is characterized by randomness in time and uncertainty in space [1]. ANSP is an essential driving factor causing the systemic pollution of the environment and its effects on the climate, directly threatening agriculture’s sustainable development and human health and safety. From a global perspective, the Earth’s surface has been affected by ANSP. Among the 1.2 billion hm^2^ degraded cultivated lands worldwide, about 12% result from ANSP. Springmann et al. (2018) suggested that during the period from 2010 to 2050, if no changes have been made to agricultural technology and institutional constraints, then the degradation of cultivated land from ANSP may increase by 50% to 90%, thereby exceeding the safe tolerance limit of the Earth [2], which will seriously hinder the United Nations sustainable development goals. Therefore, there is a growing concern about reducing the ecological environment’s systemic pollution, particularly ANSP.

Preventing ANSP has always been a hot topic in academic research. Institutional influence theory holds that the institution, as a behavioral rule, can constrain and guide the production behavior of the actors in the institutional environment, regardless of whether such behaviors and practices are efficient [3]. That is, during agricultural production, farmers not only consider the benefits of the agricultural output but also obey the “legitimacy mechanism” of ecological environment protection and adopt greener production behaviors, thereby inhibiting the aggravation of ANSP. The existing literature has shown that institutional regulations on ANSP focus on correcting externalities, and the methods include taxation, subsidies, and agreements. M Skidmore et al. (2021) found that participation in a nutrient management plan reduced ammonia and phosphorus use by 0.05 mg and 0.025 mg L, respectively, from 2010 to 2020 [4]. In a study in Nigeria, S Egbetokun et al. (2019) suggested that when pollution is effectively suppressed by social, political, and economic factors, the economy will reverse the inverted U-shaped environmental Kuznets curve [5]. The natural environment impact theory holds that the natural ecosystem reshaped by organized human activities has a profound impact on the self-sustainment and anti-interference ability of the ecosystem in the regional ecological process, which affects the regional environmental quality [6]. In a literature review conducted by PH Raven et al. (2021), the increasing agricultural intensification since World War II has included large-scale AP, monoculture, the increasing use of pesticides and fertilizers, and the elimination of spreading hedges and other fragmented wildlife habitats; all of these practices are destructive to pests and other types of biodiversity near fields. For example, the tallgrass prairies of central North America once stretched from Manitoba to northern Texas, covering an area of approximately 60 million hectares. With less than a tenth of this ecosystem left, the insect fauna of the grasslands is experiencing higher rates of loss than other species [7]. In addition, the cognitive influence theory holds that individual activities are determined by interactions among cognition, behavior, and the external environment [8]. That is, farmers’ awareness of specific behaviors directly determines their behavioral capabilities by affecting their willingness to participate. In Pakistan, good attitudes toward the use of improved grasslands arise from belief, and farmers were strongly encouraged to use enhanced plains by fellow farmers and friends, who showed a high degree of motivation to follow instructions [9].

The theory of environmental regulation (ER) is essential to the view of institutional influence. This theory holds that environmental pollution has negative externalities, and the government must use tangible institutions or intangible consciousness to restrict the actors to protect the ecological environment [10]. Since the theory was put forward, widespread attention from society has been received, particularly its outstanding contribution to preventing and controlling pollution in the industrial field. Based on the historical development of human beings, Bielecki et al. (2020) discussed the effect of various power production methods on human health and guided the development of clean energy [11]. Then, the theory was extended to the agricultural field. ER has a limited effect on preventing ANSP. For example, since 2000, China has promulgated more than 40 central and local regulations on the prevention and control of agricultural pollution (AP), and the No. 1 Central Document has continued to focus on AP over the years. However, the “China Ecological and Environmental Bulletin (2021)” pointed out that, in 2021, the fertilizer utilization rate of China’s three major grain crops, namely, rice, corn, and wheat, is only 40.2% lower than in European and American countries, which is 71.3%. The increasingly perfect environmental laws and regulations provide a basis for rational, legal, and orderly public participation in environmental protection (EP).

Moreover, the Korean Ministry of Environment aims to prevent non-point source (NPS) pollution and improve soil water management by expanding NPS priority management areas with soil loss reduction to decrease suspended solids. However, the law lacks comprehensive and detailed standards for the management of NPS priority management areas; thus, the effectiveness of the law’s implementation is unclear [12]. The advent of the digital age provides a new way toward ER. Based on the Digital Pakistan Initiative, the Pakistani government uses technology to promote social well-being in the country and further enhances ecological sustainability over time. HA Nizam et al. (2020) suggested that information and communications technology (ICT) penetration significantly reduces energy demand, where ICT infrastructure can play an essential role in reducing carbon emissions [13].

ER is based on the notion that the characteristics of “government-led” initiatives and the strong inertia of its governance model are represented by “public participation”. Although public participation is regarded as a critical factor in EP, based on a survey in rural China, only 30% of households participated in EP actions, and the contribution rate was 10%, compared with the contribution of government departments and non-governmental organizations to this topic, which reaches 30% [14]. Policy feedback theory provides an important perspective for understanding the process mechanism by which policy shapes public behavioral preferences, which proposes that this approach is essential in shaping political subsystems and environments. As an important participant in the political subsystem, the public’s cognition, attitude, and behavioral preferences will be affected by past and current policies [15].

In addition, a possible explanation for the dilemma of “government initiative, public inaction” under ER is provided. First, the cognitive behavioral theory holds that the subject’s behavioral cognition and attitude determine the subject’s motivation [8]. Farmers are the direct subjects of the control of ANSP, and their cognition of environmental rules is the basis for their participation in government. However, in real rural society, the living space of farmers is relatively closed, and the channels for farmers to obtain information are relatively narrow. Thus, “information asymmetry” is a severe problem. That is, the asymmetry between government ER and the information perception of farmers leads to the ex ante “adverse selection” with the continuous increase in ANSP. Another problem is the level of attitude and behavioral preferences. Neoclassical economics holds that farmers, as producers, are “rational economic people” who pursue profit maximization. Based on the closed living and production environment, an “acquaintance society” relationship is established among farmers based on human feelings, expression, etiquette, and customs, and the “acquaintance society” relationship in rural society has led to the existence of “mutual shelter” behavioral features among farmers [16]. Hence, with ER as a non-competitive and non-exclusive public goods supply, in practice, a farmer often uses the regulatory response behavior of other farmers to obtain public goods, such as a non-exclusive and non-competitive goods ecological environment without a bill to pay [17]. Although farmers perceive the relevant content of the government ER, they often do not respond to ER based on their interests, and rural society, with an “acquaintance society” structure, lacks external supervision. The third is the level of behavioral incentives. Regional environmental pollution data and farmers’ green production data, used as the basis for incentives, have always been characterized by “fragmentation”, “localization”, and “massiveness” [18]. The long-standing dual economic structure has divided the urban and rural production factor markets, and the rural digital infrastructure is relatively weak. Therefore, the transaction costs of financial organizations providing services to rural households are higher than those of cities. Apart from the low enthusiasm of financial service organizations in providing financial assistance to rural areas [19], the government’s use of incentives, particularly when based on digital financial services, is often prone to problems such as incentive lag, resource misallocation, and shortages, resulting in a limited effect on preventing ANSP.

In recent years, rural digitalization has developed rapidly. The “Comprehensive Survey and Research Report on Rural Revitalization in China 2021” shows that nearly 90% of rural households in China have at least one smartphone, and nearly 20% of villages have achieved “every household” broadband. Therefore, digitalization with communication technology as a key element has broken through the limitations of the geographical boundaries of traditional rural society and created a “digital virtual community” that spans between cities and villages. Therefore, this article attempts to demonstrate how to break the three above-mentioned dilemmas from the perspective of digitization. First, the “algorithmic control” behind the digital economy constitutes the path dependence of farmers’ information acquisition and perception of rules. Second, the digital empowerment of agricultural production enables the precise management of water, fertilizers, pests, weeds, and soil, as well as the accurate rationing of agricultural inputs, allowing farmers to monitor the behavior of other farmers in response to ER.

Moreover, the digital-based “virtual acquaintance society” reconstructs the internal relationship between villagers and the constraints of public opinion and addresses the long-term challenges of farmers participating in the “free-riding” dilemma. Third, artificial intelligence, cloud computing, big data, and other technologies based on digital technology have improved the efficiency and coverage of financial services in rural areas [20], empowered the incentive mechanism under ER, and solved the problems of the government’s lag in incentives, resource misallocation, and shortages. The effect of ER on the reduction in ANSP needs further analysis. That is, making digitalization an essential engine for reducing ANSP has become a critical issue.

This paper constructs a theoretical analysis framework for preventing ANSP and exploring the effect of ER following digitization using the panel data of 30 provinces from 2010 to 2020 in rural China with geographic detector tools. From the perspective of spatial stratification heterogeneity, the driving effect of ER on the reduction in ANSP is detected, the comprehensive impact of digitization and ER is explored, and the combination of ER and digitalization is explained in detail to provide a scientific-theoretical basis and practical guidance for formulating targeted and differentiated measures.

Following previous studies, this study greatly contributes from three aspects. First, the impact of ER on the reduction in ANSP in the spatial dimension is discussed with consideration of the impact of digital superposition and its spatiotemporal characteristics. The research on the effect of ER has been focused on traditional regression relations, such as ordinary least squares and fixed-effect models, following the assumptions that spatial heterogeneity does not exist, and lacks the exploration of the non-stationarity of space and time. Second, the geographic detector analysis method for research is used to identify the problem of spatial stratification heterogeneity, in which the variance within the stratum is smaller than the variance between the strata, thereby addressing the assumptions of data homoscedasticity and the processing of categorical variables by traditional statistical methods. Our research advances a more comprehensive understanding of the literature on the factor interaction between ER and digitization as a critical yet poorly identified path that links to the reduction in ANSP. Third, with regard to the method used, the practical interpretation of spatial factors is emphasized with application scalability. Our study complements the empirical spatial evidence of ER and digitization across changes in spatial positions.

## 2. Hypothesis Development

### 2.1. Prevention of ANSP with ER

As a public social product, the agricultural ecological environment is non-exclusive and non-competitive. Farmers often adopt “free-rider behavior” during agricultural EP. Farmers are usually unwilling to pay for such public goods in the absence of economic compensation and incentives, leading to the continued occurrence of the “tragedy of the commons” [21]. Therefore, the “visible hand” of the government must restrict the behavior of farmers by formulating three types of ER, incentive, guidance, and restriction, to achieve the established goal of reducing ANSP. Neoclassical economics holds that farmers, as producers, are “rational economic people” who pursue profit maximization.

Regarding incentives, whether farmers adopt green production methods depends on AP’s costs and expected benefits [22]. When farmers realize that the government’s use of incentives, such as economic compensation, has significantly reduced the cost of green production methods or increased expected benefits, they tend to comply with ER and adopt agricultural green production behaviors. For example, local governments use various financial subsidies to promote the resource utilization of agricultural waste, change the direction of the use of monetary donations, and shift price subsidies primarily used for the purchase and sale of chemical fertilizers and pesticides to support the research and development of agricultural green production technologies. Incentive subsidies for ecological farming activities [23] guide farmers from different perspectives to shift toward pro-environmental production methods. With regard to guidance methods, the government promotes the dissemination of agricultural green production information and farmers’ recognition of ER through publicity on EP, guidance, and training on agricultural green production technologies [24]. Farmers with a higher degree of ER recognition likely form a comprehensive understanding of EP, increasing the possibility of their adoption of green production behaviors to reduce ANSP. With regard to restrictive measures, local governments have proposed control measures for different types of pollution sources, such as chemical fertilizers and pesticides, by issuing relevant documents or formulating strict pollution control regulations (such as the registration system for chemical fertilizers and pesticides, delineating areas where breeding is prohibited and restricted), controlling ANSP at the source. Farmers who deviate from the set goals will face administrative penalties such as fines. Hence, farmers with a strong awareness of ER often weigh the cost of violations before implementing the behavior and adopt agricultural green production behaviors because of their economic rationality to avoid losses [25], thereby effectively preventing ANSP. Therefore, Hypothesis 1 can be obtained.

**Hypothesis** **1.**
*ER has a significant positive effect on preventing ANSP.*


### 2.2. Prevention of ANSP Following Digitalization

Digitalization uses data as the core production factor. By applying digital technologies (e.g., the Internet of Things and big data), new kinetic energy is given to production factors, such as the infrastructure, technology, and capital required for green agricultural transformation, thereby reducing ANSP. First, digital agriculture combines AP equipment, such as agricultural machinery and agricultural conditions, with digital technologies, such as the Internet of Things, remote sensing, and drones. It installs navigation devices (e.g., geographic information systems) and application devices (e.g., variable water and fertilizer control systems) [26]. The precise management of water and fertilizer, pests, weeds, and soil can be carried out to avoid the excessive use of chemical fertilizers and pesticides. Second, green finance is an economic activity that supports environmental improvement, climate change response, resource conservation, and efficient use [27], while digital financial inclusion emphasizes equalizing opportunities for all sectors of society to obtain financial services [28]. Green finance emphasizes quality, whereas digital financial inclusion reflects equality and efficiency, which indicates that with the support of digital technology, financial institutions will provide farmers with more efficient, fairer, and greener financial services, thereby improving the financial accessibility to rural agriculture farmers [29]. From the perspective of production, the coordinated development of inclusive finance and green finance provides financing channels and financing facilities for the green R&D and green production behaviors of agricultural enterprises or farmers by promoting the innovation of green financial products and improving the efficiency of financial services, which is conducive to the transformation and upgrading of green agricultural production technology or adoption, and then the prevention of ANSP [30]. From the perspective of the transaction side, supply-and-demand theory holds that the demand has a pushing and pulling effect on the supply. In contrast, consumer finance and credit have a significant stimulating effect on domestic demand [31], indicating that the “Green Financing Statistics System” issued by the China Insurance Regulatory Commission includes green agricultural product consumption, ecotourism, and health tourism. In addition, green standards such as “three products and one standard” in the agricultural field will become an important starting point for financial support and for the development of green agriculture. The green supply of agriculture is stimulated to guide the green production behavior of farmers, thereby preventing ANSP [27]. Therefore, Hypothesis 2 is proposed.

**Hypothesis** **2.**
*Digitalization significantly positively affects the reduction in ANSP.*


### 2.3. Effect of ER on Digitalization

Although the construction of the government-led ER system has been continuously improved, in practice, farmers have the dilemma of “participation failure”. First, the asymmetry between the government ER and the information perception of farmers leads to the ex ante “adverse selection” in the production process of farmers. Second, the policy response behavior of farmers during AP, coupled with the “mutual shelter” behavior by the “acquaintance society” relationship structure among farmers, lacks monitoring, resulting in the “free-riding” behavior of farmers participating in ER. In addition, the lack of rural financial services has resulted in the failure of incentive means, and farmers’ policy response behaviors lack enthusiasm. These dilemmas can be solved with digital technology. Digitalization with communication technology as a critical element opens the relatively closed rural social environment and becomes an essential engine for farmers to interact with the outside world. Based on various new media platforms, social information can be rapidly disseminated through point-to-point and face-to-face communication, interpersonal communication, and mass communication. Moreover, the digital economy is a set of sophisticated hidden-power governance techniques, which are directly manifested as “algorithmic control”, that is, through algorithms, to analyze public psychological preferences and implement agenda setting for information flow for users [29]. This controls the information content users are exposed to and constitutes the path dependence of farmers’ rule acquisition and perception.

Second, the digital empowerment of AP can be used to carry out the precise management of water, fertilizer, pests, weeds, and soil, as well as the accurate monitoring of farmers’ agricultural inputs [28], achieving the behavioral monitoring of farmers’ responses to ER. However, the established rural acquaintance society was replicated on the digital platform to form a “virtual acquaintance society”. In the “virtual acquaintance society”, the village’s public-opinion-based restraint mechanism is gradually being demonstrated through Internet practices [32]. Given the lack of digital communication, the characteristics of face-to-face communication often encourage villagers to express justice. Compared with the traditional acquaintance society, the public opinion of the virtual acquaintance society spreads faster and more effectively. Therefore, digitalization has reconstructed the internal relationship between villagers and the constraints of public opinion, breaking the mutual cover-up behavior among farmers in the traditional “acquaintance society” [18] and overcoming the long-term challenge of farmers participating in the “free-rider” dilemma.

On the contrary, given the failure of incentives due to the lack or high costs of rural financial services, digitization can utilize technologies such as the Internet, big data, and blockchain to expand relevant data to behavioral data, such as text data. For unstructured data such as picture data, digital technologies such as artificial intelligence and cloud computing are introduced to create a financial data center that automatically collects and analyses data and forms charts or analysis reports to process fragmented and massive data at low cost, reducing information integration costs and transaction costs [33,34], improving the efficiency and coverage of financial services in rural areas, and then empowering incentives under government ER to make them more comprehensive, accurate, green, and efficient. Therefore, Hypothesis 3 is obtained.

**Hypothesis** **3.**
*Digital empowerment of government ER has a significant positive effect on the reduction in ANSP.*


## 3. Data, Variables, and Model

### 3.1. Data Sources

This study used the panel data of 30 provincial-level administrative regions in China from 2010 to 2020 as the empirical sample (given the availability of data, Tibet, Hong Kong, Macao, and areas of Taiwan are not included). Among them, the raw data on chemical fertilizers, pesticides, agricultural mulch, agricultural diesel, and agricultural land area required to measure the intensity of ANSP were obtained from the statistical yearbooks of the 30 provincial-level administrative regions. The number of enacted ERs is from the Law of Peking University Database, whereas government work reports “Agricultural Pollution Control”, “Agricultural Ecology”, and “Rural Ecology” can be accessed from local (municipal) government information networks. The investment magnitude in agricultural EP was obtained from the “China Fiscal Yearbook”, “China Environmental Statistical Yearbook”, and the financial and ecological statistical yearbooks of provincial administrative regions. The input of innovative digital equipment in agricultural sectors such as planting and animal husbandry was obtained from the “China Agricultural Machinery Industry Statistical Yearbook”. The rural Internet broadband penetration rate is from the “China Regional Economic Statistical Yearbook”. The green finance index is from the Green Finance International Research Institute of the Central University of Finance and Economics and the Digital Finance Research Center of Peking University. The digital financial inclusion index can be obtained from the Digital Finance Research Center of Peking University. We also accessed the primary geographic information data from the National Geographic Information Center database. For missing data, interpolation was used for imputation.

### 3.2. Variables

#### 3.2.1. Explained Factors

The measurement of ANSP intensity was carried out as follows. Considering that the mechanism of ANSP is complex, random, and uncertain, that is, as long as chemical fertilizers, pesticides, and other agricultural materials are used in agricultural production, water resources and soil resources will be more or less affected by rainfall, irrigation, and other channels (e.g., pollution frequency). Consequently, ANSP becomes difficult to quantify from the intensity of emissions. Accordingly, the measurement method of ANSP in this study is based on the measurement method of Jiang Song. The United Nations Human Development Index weighting method was adopted. The average input amounts of chemical elements such as fertilizers, pesticides, agricultural film, and diesel are used to measure pollution [35]. Given the differences in the natural conditions and economic development levels of various provinces, measurements of the average input amounts are not comparable. Therefore, the intensity of ANSP, the ratio of the average input amount to the area of agricultural land, the specific formula is as follows:(1)$$PI_{\mathrm{it}}=\frac{0.25\times (CF_{\mathrm{it}}+PE_{\mathrm{it}}+MF_{\mathrm{it}}+DE_{\mathrm{it}})}{S}$$
where *PI* is the ANSP intensity in the *t*-th province in the *i*-th year, which are numerical quantities. *CF*, *PE*, *MF*, and *DE* are the application amounts of chemical fertilizers, pesticides, agricultural mulch film, and agricultural diesel (10,000 tons), respectively, and *S* is the area of agricultural land (10,000 ha). The value 0.25 means the weighted average mean of *CF*_it_, *PE*_it_, *MF*_it_, and *DE*_it_.

Figure 1 shows the spatial distribution of ANSP in 2010 and 2020. There is spatial heterogeneity, with considerable spatial differentiation in ANSP across China. The spatial differentiation characteristics of ANSP are as follows. The ANSP intensity decreased from the southeast coast to the northwest inland in 2010, but the spatial distribution was too loose to emerge as a prominent characteristic in 2020.

#### 3.2.2. Explanatory Factors

For ER factors, government investment in environmental protection is an index to measure the incentive means of ecological regulation [36]. Thus, the proportion of agricultural and EP investment in GDP was used. Government attention is an index to measure the guiding means of ER [5], the measure of which is the number of “agricultural pollution control”, “agricultural ecology”, and “rural ecology” government work reports. Rules and regulations are the index to measure the restrictive means of ER [5,36], the measure of which is the number of regional ERs promulgated.

Regarding digital factors, digital infrastructure construction is used to measure the level of rural digital building [37], and the penetration rate of rural Internet broadband is used as the measurement item. Digital application expresses the application of digital technology to agricultural production activities [37] using planting and animal husbandry. The proportion of investment in innovative digital equipment in the agricultural sector and other agricultural industries in the total investment is used as a measurement item. Green inclusive finance integrates green finance and inclusive digital finance. It refers to the support of digital technology to enable all sectors of society to obtain approval for environmental improvement, the equalization of opportunities for climate change and resource conservation, and the efficient use of economic activities [29], using the interaction between the green financial index and the digital financial inclusion index as the measurement item.

The above-mentioned variables are numerical quantities, as shown in Table 1.

### 3.3. Model Selection

The geographic detector method is a statistical method based on the heterogeneity of a particular attribute among different geographical fields to explore the driving effects of various explanatory factors on the factors to be explained. Its advantages are numerous. First, the model has no linear hypothesis, which can better overcome the multicollinearity problem and endogeneity problem. Second, we should pay attention to the objective fact that the “sample” is located in the geographical space, emphasize the spatial heterogeneity of the intra-layer variance less than the inter-layer variance, and reveal the driving factors behind it based on this spatial heterogeneity. Third, the impact of two explanatory factors’ interaction on the explained factor is determined. The identification method of exchange in the regression model is multiplicative, but the interaction between two elements is not necessarily multiplicative. The geographical detector can calculate and compare the q value of every single factor as well as the q value after the superposition of the two elements, determining whether there is an interaction between the two factors and the strength and direction of the interaction [38,39]. Therefore, this study primarily used factor detection. Factor interaction detection was used to test the driving effects of ER, digitalization, and their interaction on the prevention of ANSP.

#### 3.3.1. Factor Detector

This method is primarily used to detect the spatial differentiation of the explained factor *Y* and that based on *Y*. The extent to which the explanatory factor *X* explains the explained factor *Y* is calculated by the following formula.
(2)$$q=1-\frac{\sum_{h=1}^{L}N_{h}\sigma_{h}^{2}}{N\sigma^{2}}$$

In the formula, the q value measures differentiation and factor detection. The larger the q value, the stronger the explanatory power of the explanatory factor *X* to the explained factor *Y*. The h is the number of sub-regions of factor *X*, and *N* represents the space of the entire research area, that is, the total number of units (provinces in this study). *N*_h_ represents the sample size of sub-region h, and *σ*_h_ indicates the sample total variance and variance of sub-region h.

#### 3.3.2. Factor Interaction Detector

Interaction detection is based on Y-based spatial differentiation, identifying whether the interaction between the explanatory factor X_1_ and the explanatory factor X_2_ will increase or decrease the explanatory power of the explained factor Y. The calculation steps are as follows. First, the explanatory factor X is calculated. For the q value of the defined factor Y: q (X), the q value can be obtained when they interact (i.e., q(X_1_ ∩ X_2_)). Secondly, for q(X_1_), q(X_2_), and q (X_1_ ∩ X_2_), the judgment criteria are shown in Table 2.

## 4. Results and Analysis

### 4.1. Influencing Factors of ANSP

#### 4.1.1. Driving Effect of ER

Figure 2 shows the spatial difference in ER in among 30 provinces in China in 2020. Government investment in environmental protection has been increasing from the southeast coast to the northwest inland, with an obvious trend of extension, whereas government attention has decreased alternately from the southeast coast to the northwest inland, with an obvious trend of extension. For rules and regulations, the legal norms have been declining alternately from the southeast coast to the northwest inland and have been declining continuously from the east to the west, with an obvious trend of extension. Hence, Hypothesis 1 can be verified.

Table 3 reports the driving effect of ER on preventing and controlling ANSP from the perspective of factor comparison, and the driving force is 0.322. In particular, government attention has the most potent driving force in reducing ANSP, with a driving force of 0.391, followed by legal regulation, with a driving force of 0.382, and the driving force of government investment in protection and governance is 0.192. Therefore, the guiding, restrictive, and incentive means of ER significantly drive the prevention of ANSP, and the order decreases. Farmers with a strong awareness of ER often weigh the cost of violation before implementing their behavior and are driven to adopt agricultural green production behavior by their economic rationality to avoid losses, thereby effectively preventing ANSP. From 2010 to 2020, the driving force of restrictive measures to prevent ANSP ranged from 0.407 to 0.384 to 0.356, showing a “U”-shaped change trend, whereas the driving force of incentive measures ranged from 0.139 to 0.139. From 0.281 to 0.156, an “inverted U”-shaped movement is observed. The leading driving force increases from 0.342 to 0.489, showing a “constantly rising” trend. It is reasonable to deduce that ER compensates for the agricultural operation cost to force peasants to innovate in green production technology, thus preventing ANSP.

Based on the interpretation of the National Development and Reform Commission, the east, middle, and west divisions of China are policy divisions, not administrative divisions or geographical definitions. Therefore, the east refers to the provinces and cities that have implemented the coastal opening policy earlier and have a high level of economic development, including Beijing, Tianjin, Hebei, Liaoning, Shanghai, Jiangsu, Zhejiang, Fujian, Shandong, Guangdong, and Hainan. The central part refers to the economically underdeveloped areas, including Shanxi, Inner Mongolia, Jilin, Heilongjiang, Anhui, Jiangxi, Henan, Hubei, and Hunan. The west refers to the underdeveloped western regions, including Sichuan, Guizhou, Yunnan, Tibet, Shaanxi, Gansu, Qinghai, Ningxia, Xinjiang, and Guangxi.

Table 4 expresses the driving effect of ER on preventing ANSP from the perspective of horizontal and local comparisons. The results show significant differences in the core driving factors in the eastern, central, and western regions. In particular, the east part primarily provides “guiding means (0.281) > incentive means (0.265) > restrictive means (0.214)”. In the central region, the performance is “guiding means (0.362) > restrictive means (0.354) > incentive means (0.291)”. In the western part, the order is “restrictive means (0.412) > incentive means (0.375) > guiding means (0.369)”. Notably, at the regional level, the driving effect of ER on the prevention of ANSP from 2010 to 2020 is negatively correlated with the level of regional economic development.

#### 4.1.2. Driving Effect of Digitalization

Figure 3 shows the spatial differences in digitalization among China’s 30 provinces in 2020. The level of digital infrastructure construction is declining from the southeast coast to the northwest inland, but it is relatively scattered, with no obvious distribution trend, and its spatial differentiation is large. Rural digital application is declining from the southeast coast to the northwest inland, and the distribution trend is relatively obvious. Green inclusive finance is also in a downward trend from the southeast coast to the northwest inland, with an obvious extension trend. Hypothesis 2 can be verified.

Table 5 reports the driving effect of digitalization on reducing ANSP from the perspective of factor comparison. The driving force is 0.394, which is more significant than the driving effect of ER factors on ANSP. In particular, green inclusive finance has the most potent driving effect on the prevention of ANSP, with a driving force of 0.429, followed by digital application, with a driving force of 0.394. The driving force of digital infrastructure construction is 0.359. Therefore, digitalization has given new momentum to production factors such as the infrastructure, technology, and capital required for the transformation of agricultural greening and has made outstanding contributions to the prevention of ANSP. Digitalization uses data as the core production factor. The application of digital technologies such as the Internet of Things and big data has given new impetus for production factors such as the infrastructure, technology, and capital needed for the transformation of agricultural greening, thus reducing agricultural non-point source pollution. From 2010 to 2020, the driving effect of digitalization on the prevention of ANSP showed a “constantly rising” trend, with inclusive finance showing the fastest rise from 0.379 to 0.429, followed by digital application, which increased from 0.366 to 0.415. Finally, digital infrastructure construction rose from 0.346 to 0.373.

Table 6 expresses the horizontal and local comparative angles and the driving effect of digitalization on the prevention of ANSP. The results show significant differences in the core driving factors of the east, middle, and west. In particular, in the eastern region, the central performance is “green inclusive finance (0.421) > digital application performance (0.359) > digital infrastructure construction (0.324)”. In the central region, the performance is “digital infrastructure construction (0.458) > green inclusive finance (0.398) > digital application performance (0.367)”. In the western region, the performance is “digital infrastructure construction (0.491) > digital application performance (0.432) > green inclusive finance (0.412)”. Notably, at the regional level, the driving effect of digital application performance and digital infrastructure construction on the prevention of ANSP is negatively correlated with the level of economic development from 2010 to 2020, whereas the driving role of green inclusive finance is positively correlated with the level of economic development. In this vein, it is reasonable to anticipate that digitalization, which is scientific and technological progress leading to innovation, can prevent ANSP with an update for the green transformation of agricultural production.

### 4.2. Interaction of Influencing Factors of ANSP

The six explanatory factors are defined to further explore changes in the explanatory power of the prevention of ANSP when different detection factors interact. The six explanatory factors are defined as follows: X_1,_ ER; X_2,_ government attention; X_3_, government; EP investment; X_4,_ digital application performance; X_5_, digital infrastructure construction; and X_6_, green inclusive finance (green finance × digital inclusive finance). As shown in Table 7, ER and digitalization are not independent of each other, but a relatively close relationship is observed. That is, the interaction between ER and digitalization is greater than the driving force of a single element on the prevention of ANSP, indicating that from the perspective of differentiation, the reduction in ANSP results from multiple influencing factors, namely, a significant positive effect of the digital empowerment of government ER.

From the perspective of the horizontal factor interaction comparison, the interaction between ER and digital infrastructure construction has the most substantial explanatory power on the prevention of ANSP, with a driving force of 0.863. In addition, the interaction between government attention and digital infrastructure construction is 0.815, indicating that the restrictive and guiding means under the regulation of the digital empowerment environment constitute the path dependence of farmers’ rule acquisition and perception, breaking the mutual sheltering behavior between farmers in the traditional “acquaintance society” and overcoming the difficulty of farmers participating in the “free ride” dilemma. It is a long-term challenge that positively affects the prevention of ANSP. The driving forces of the interaction between government protection and governance investment, digital infrastructure construction, and green inclusive finance are 0.742 and 0.798, respectively, which shows that digital technology empowers the horizontal adjustment of economic incentives under government ER, which verifies Hypothesis 3. It is comprehensive, accurate, green, and efficient, and it has a significant driving effect on the prevention of ANSP. The exchange of inclusive finance shows an increasing trend, among which the interaction of government protection, governance investment, and digital infrastructure construction has the fastest growth rate, with an increase of 0.163.

## 5. Conclusions 

By constructing the analysis framework of the reduction in ANSP with ER, this study analyzed the internal mechanism by which ER, digitalization, and their interaction affect the prevention of ANSP. The panel data of provinces (given data availability, excluding Tibet, Hong Kong, Macao, and Taiwan regions), based on spatial heterogeneity, were empirically tested using geographic detectors. The results show that ER has a driving effect on reducing ANSP, and the driving force is 0.322, but this effect is obtained from constraints of incentive (0.192), guiding (0.391), and restrictive (0.382) means on farmers’ behavior under the government ER. Compared with the national sample, the core driving force in the eastern and central regions is guiding means (0.281 in the eastern part and 0.362 in the western area), whereas the core driving force in the western region is restrictive means (0.412).

Compared with ER, the driving effect of digitalization on the prevention of ANSP is more significant, and the driving force is 0.394. Digitalization has given new momentum to production factors such as the infrastructure (0.359), technology (0.394), and capital (0.429) required for green agricultural transformation. Regarding the national sample, the core driving factor in the eastern region is green inclusive finance (0.421), and the core driving factor in the central and western regions is digital infrastructure construction (0.458 in the central area and 0.491 in the western part). Further factor interaction detection indicates that the interaction between ER and digitalization has a significant driving effect on preventing ANSP because digitalization constitutes the path dependence of farmers’ rule acquisition and perception, which addresses the “free rider” dilemma of farmers’ participation (0.863). Furthermore, digital technology empowers incentives under government ER (0.770), making it more comprehensive, accurate, green, and efficient.

## 6. Policy Recommendations

The reduction in ANSP is not a long-term goal but a substantive process to achieve results. Combining the above-mentioned conclusions, this paper draws the following policy implications.

First, the role of the government as a “guide” in the green development of agriculture is suggested to be given. Initiatives consisting of strengthening the top-level scientific design, formulating green agricultural development paths, and determining critical areas and areas for the prevention of ANSP can improve laws and regulations for agricultural ecological civilization, standardize AP, and cultivate a group of specific regions that are active in prevention and control work. By increasing the financial inclination of agricultural materials enterprises for green agricultural production technology innovation and playing the “leadership” role of agricultural materials enterprises in technological innovation through industry benchmarking and model establishment, the government’s institutional advantages of “concentrating power to do big things” and strengthening the construction of rural infrastructure can be realized, particularly the establishment of an agricultural production ecological data collection platform. This platform can systematically and quantitatively evaluate the current status of AP ecology and make timely improvements and adjustments. Promoting the environmental construction of AP has become the core driving force for the green development of agriculture.

Second, the channels and paths for farmers to participate in preventing ANSP could be broadened and optimized. On the one hand, policymakers could actively improve laws and regulations on the prevention of ANSP, strictly implement the government information disclosure system, and increase the construction of ANSP information disclosure systems in villages and towns. On the other hand, a combination of legal publicity, vocational training, and roadshows could be adopted, making full use of information dissemination channels such as the Internet to popularize initiatives. By promoting the knowledge of green AP, farmers’ EP awareness is improved, and the regulatory role of farmers is paid attention to.

Third, policymakers could increase digital construction in rural areas and consolidate the underlying structure of Digital China. Using the digital construction plan, an implementation mechanism guided by government funding can be formed for the extensive participation of social capital and strict supervision by social groups, ensuring reasonable resource investment. The establishment of a universal service compensation mechanism for rural telecommunications would support the construction of optical fiber networks and 5G base stations in villages and towns, achieve the “same network and same speed” in rural cities, narrow the “digital divide” between urban and rural areas, and unblock the application channels of digital AP technology and AP. Then, the development of high-end technologies such as big data and blockchain and their application to AP can be promoted to increase the coverage of digital financial inclusive services and provide digital financial “pilot fault tolerance” space under the premise of maintaining the bottom line. Furthermore, the green effect of digital finance needs to be realized to help the green transformation and upgrading of agricultural economic development and to prevent ANSP.

Fourth, for the eastern and central regions, central planners’ cross-sectoral and cross-local coordination role in regional environmental policies utilizing guidance, regional planning, policy documents, and other ways to strengthen sectoral and regional cooperation could be improved to promote cross-regional comprehensive governance. In addition, a policy that supports the publicity of inclusive digital finance in rural areas effectively can improve farmers’ understanding and acceptance of inclusive digital finance, as well as financial availability, and ensure that inclusive digital finance and green finance play a better role in environmental protection. For the western region, the support of relevant departments for the construction of rural digital infrastructure in remote western areas and the publicity and demonstration of digital rural application scenarios need to be strengthened, and rural residents’ awareness of and willingness to use digital technology need to be enhanced.

## Figures and Tables

**Figure 1 ijerph-20-04396-f001:**
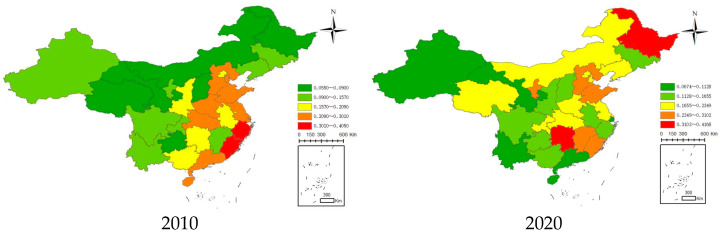
Spatial distribution of ANSP in 2010 and 2020.

**Figure 2 ijerph-20-04396-f002:**
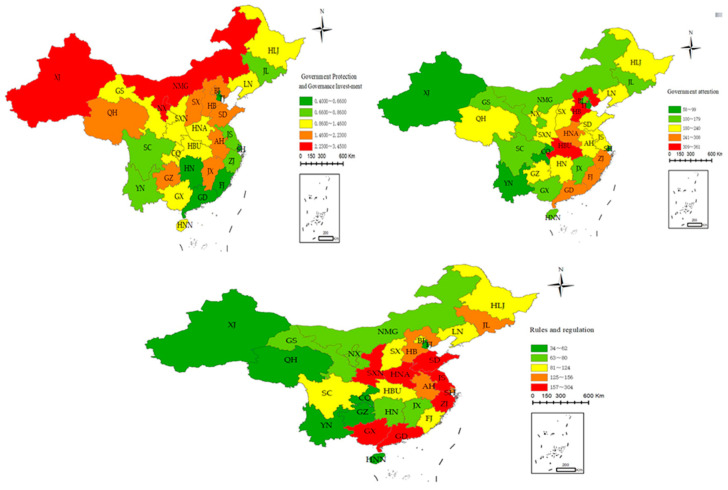
The spatial distribution of ER. Note: The upper left, upper right, and bottom graphs reflect the spatial distributions of government investment in environmental protection, government attention, and rules and regulations, respectively.

**Figure 3 ijerph-20-04396-f003:**
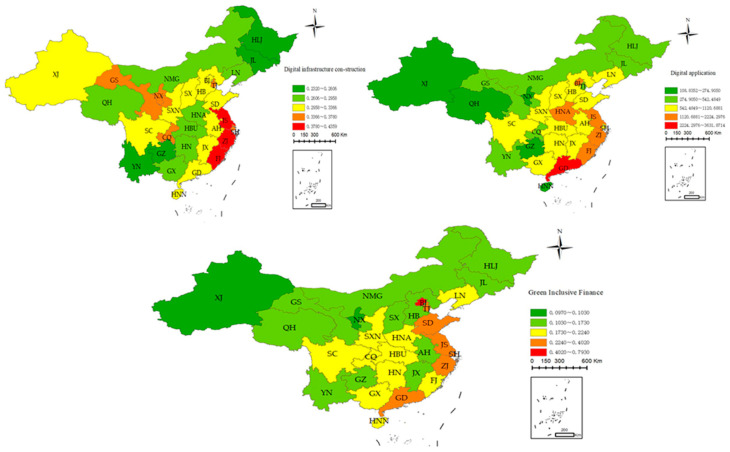
The spatial distribution of digitalization. Note: The upper left, upper right, and bottom graphs reflect the spatial distributions of digital infrastructure construction, digital application, and green inclusive finance, respectively.

**Table 1 ijerph-20-04396-t001:** Variable description and descriptive statistics.

Factor	Meaning	Average	Standard Deviation
ANSP	ANSP intensity	0.181	0.084
Government investment in environmental protection	Agricultural and environmental protection investment as a percentage of GDP (%)	0.352	0.545
Government attention	Several government work reports on “Agricultural Pollution Control”, “Agricultural Ecology”, and “Rural Ecology”.	208	331.212
Rules and regulations	Number of enacted environmental laws (ministries)	135.689	126.776
Digital infrastructure construction	Rural Internet broadband penetration rate (%)	0.203	0.099
Digital application	Agricultural sectors such as planting and animal husbandry in total investment (%)	0.312	0.574
Green inclusive finance	Green finance index	0.181	0.111
	Digital financial inclusion index	270.885	721.724

**Table 2 ijerph-20-04396-t002:** Types of interactions.

Judgment Basis	Interaction Type
q(X_1_ ∩ X_2_) < min [q(X_1_), q(X_2_)]	Non-linear attenuation
min [q(X_1_), q (X_2_)] < q(X_1_ ∩ X_2_) < max [q(X_1_), q(X_2_)]	One-factor non-linear attenuation
q(X_1_ ∩ X_2_) > max [q(X_1_), q (X_2_)]	Two-factor enhancement
q(X_1_ ∩ X_2_) = q(X_1_) + q(X_2_)	Independent relationship
q(X_1_ ∩ X_2_) > q(X_1_) + q(X_2_)	Non-linear enhancement

**Table 3 ijerph-20-04396-t003:** Detection results of global ER drivers.

Factor	2010		2015		2020		2010–2020	
	q Value	Ranking	q Value	Ranking	q Value	Ranking	q Value	Ranking
Government investment in environmental protection	0.139 *	3	0.281 *	3	0.156 *	3	0.192 *	3
Government attention	0.342 **	2	0.343 **	2	0.489 **	1	0.391 ***	1
Rules and regulations	0.407 **	1	0.384 ***	1	0.356 **	2	0.382 **	2
Average contribution rate	0.296		0.336		0.334		0.322	

Note: *, **, and *** represent significance levels of 10 %, 5%, and 1%, respectively. “Factor Aver-age Contribution Rate” represents the mean value of the above detection factors.

**Table 4 ijerph-20-04396-t004:** Detection results of local ER drivers.

Factor	East		Middle		West	
	q Value	Ranking	q Value	Ranking	q Value	Ranking
Government investment in environmental protection	0.265 **	2	0.291 **	3	0.375 **	2
Government attention	0.281 ***	1	0.362 **	1	0.369 **	3
Strength of regulation	0.214 **	3	0.354 **	2	0.412 ***	1
Average contribution rate	0.253		0.336		0.385	

Note: **, and *** represent significance levels of 5%, and 1%, respectively. The east, middle, and west regions refer to administrative divisions in China.

**Table 5 ijerph-20-04396-t005:** Detection results of global digital drivers from 2010 to 2020.

Factor	2010		2015		2020		2010–2020	
	q Value	Rank	q Value	Rank	q Value	Rank	q Value	Rank
Digital infrastructure construction	0.346 *	3	0.357 *	3	0.373 ***	3	0.359 ***	3
Digital application	0.366 ***	2	0.401 ***	2	0.415 **	2	0.394 ***	2
Green inclusive finance (green finance × digital inclusive finance)	0.379 **	1	0.433 ***	1	0.474 ***	1	0.429 **	1
Average contribution rate	0.364		0.397		0.421		0.394	

Note: *, **, and *** represent significance levels of 10 %, 5%, and 1%, respectively.

**Table 6 ijerph-20-04396-t006:** Detection results of local digital drivers from 2010 to 2019.

Factor	East		Middle		West	
	q Value	Rank	q Value	Rank	q Value	Rank
Digital infrastructure construction	0.324 **	3	0.458 **	1	0.491 ***	1
Digital application performance	0.359 **	2	0.367 **	3	0.432 ***	2
Green inclusive finance (green finance × digital inclusive finance)	0.421 ***	1	0.398 ***	2	0.412 ***	3
Average contribution rate	0.368		0.408		0.445	

Note: **, and *** represent significance levels of 5%, and 1%, respectively. The east, middle, and west regions refer to administrative divisions in China.

**Table 7 ijerph-20-04396-t007:** Interaction detection results of influencing factors of ANSP.

Factor Interaction	2010		2015		2020		2010–2020	
	q Value	Type	q Value	Type	q Value	Type	q Value	Type
1 ∩ 2	0.482 *	DE	0.404 *	DE	0.559	DE	0.482 *	DE
1 ∩ 3	0.635	DE	0.723 **	DE	0.856 *	DE	0.738 **	DE
1∩ 4	0.738 *	DE	0.624 *	DE	0.583	DE	0.648*	DE
1 ∩ 5	0.796 *	DE	0.650 **	DE	0.845 *	DE	0.863 **	DE
1 ∩ 6	0.832 **	DE	0.801 *	DE	0.777 *	DE	0.764	DE
2 ∩ 3	0.622 *	DE	0.701	DE	0.852 *	DE	0.725 *	DE
2 ∩ 4	0.704 *	DE	0.646 **	DE	0.579 *	DE	0.643 **	DE
2 ∩ 5	0.684 ***	DE	0.821 *	DE	0.940 **	DE	0.815 *	DE
2 ∩ 6	0.752	DE	0.799 *	DE	0.794 *	DE	0.782 *	DE
3 ∩ 4	0.527 *	DE	0.709	DE	0.833 *	DE	0.690 *	DE
3 ∩ 5	0.597 *	DE	0.839 **	DE	0.789 ***	DE	0.742 **	DE
3 ∩ 6	0.781 ***	DE	0.823	DE	0.790 *	DE	0.798 **	DE
4 ∩ 5	0.595 **	DE	0.703 **	DE	0.777 **	DE	0.692 **	DE
4 ∩ 6	0.672 **	DE	0.764	DE	0.788 **	DE	0.741 **	DE
5 ∩ 6	0.432	DE	0.762 *	DE	0.747 *	DE	0.647 *	DE

Note: *, **, and *** represent significance levels of 10%, 5%, and 1%, respectively. DE: double enhancement.

## Data Availability

Not applicable.

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
