# Peer review of "Preventing Agricultural Non-Point Source Pollution in China: The Effect of Environmental Regulation with Digitization"

_ijerph, 2023, doi:10.3390/ijerph20054396_

Round 1
Reviewer 1 Report
1. Original Submission
1.1. Recommendation
Major revision
2. Comments to Author:
Title: How to Reduce Agricultural Non-point Source Pollution:An Effect of Environmental Regulation with the Digitization.
Overview and general recommendation:
This study tried to confirm the effect of environmental regulation with the digitization to reduce agricultural non-point source pollution using geographic detector. Overall, the idea is acceptable, but the arrangement and structure of the paper was poor, and the novelty of the study needs to be improved. I do not recommend it be published at the present form, and a thorough major revision is needed. I explain my concerns in more detail below.
1. The factors selected in the study are more targeted at the prevention and control of ANSP in China. It is suggested to change the title to “ How to Reduce Agricultural Non-point Source Pollution in China:An
Effect of Environmental Regulation with the Digitization”.
2. Environmental regulation with the digitization has been studied to prevent and control ANSP, and the geographic detector has been applied to
regional planning, ecology, environmental pollution, remote sensing and other fields successfully. Therefore, the novelty of this study needs to be strengthened.
3. In Section â…¢, there is no description to the type and processing methods of the variables. It is suggested to supplement the processing methods of different types of variables (eg. numerical quantity, type quantity).
In addition, agricultural non-point source pollution has spatial heterogeneity, and the spatial distribution map of factors are not seen in the paper. Developer's literature should be given in “model selection”. For example, “Jinfeng WANG, Chengdong XU. Geodetector: Principle and prospective[J]. Acta Geographica Sinica, 2017,72(1): 116-134.”.
4. The full English name notes should be given for the first abbreviation. Specifically, P3L101-102 “ICT” and P4L194 “EP”.
5. There are some points for attention as follows:
P3L130, L132 and P4L192-193. The second “non” is missing in “non-exclusive and -competitive”.
P7L343. What is the meaning of 0.25 in the formula (1)?
P7L345. There is a space which should be removed between D and E.
P7L349 and L356.The “factor” needs to be plural.
P8L368 and P9L418. I think that “Government Protection and Governance Investment” in the first column of Table 1 and “Government Investment in Environmental Protection” of Table 3 should be the same variable.
P8L368 Table 3and P11L459 Table 6. The initial letters of variables in “Factor” are confused. It is recommended to unify.
P8L386. The “explained” of “the explained factor X” is wrong, and it should be “explanatory”.
P9L393. “S or Sh” doesn’t appear in the formula (2), it should be “s or sh”.
P10L429 and P11L459. What regions are the "East, Middle and West" in Tables 4 and 6, respectively? Are they China's three major economic belts? It is suggested to give notes.
P12L490-491. No note is given for “OF” in Table 7.
Author Response
#1 The factors selected in the study are more targeted at the prevention and control of ANSP in China. It is suggested to change the title to “ How to Reduce Agricultural Non-point Source Pollution in China:An Effect of Environmental Regulation with the Digitization”.
Reply: Thank you for your comment. The title has been changed to "Preventing Agricultural Non-point Source Pollution in China: An Effect of Environmental Regulation with Digitization".
#2 Environmental regulation with the digitization has been studied to prevent and control ANSP, and the geographic detector has been applied to regional planning, ecology, environmental pollution, remote sensing and other fields successfully. Therefore, the novelty of this study needs to be strengthened.
Reply: Thank you for your reminder. Our contribution of this manuscript has been rewritten in line 181-193.
For instance," The research on the effect of ER has been focused on the relation of traditional regression, such as ordinary least squares and fixed-effect models, following the assumptions that spatial heterogeneity does not exist, and lacks the exploration of the nonstationarity of space and time. Second, the geographic detector analysis method for research is used to identify the problem of spatial stratification heterogeneity in which the variance within the stratum is smaller than the variance between the strata, thereby addressing the assumptions of data homoscedasticity and the processing of categorical variables by traditional statistical methods. Our research advances a more comprehensive understanding of the literature on the interaction factors between ER and digitization as a critical yet poorly identified, path that links how to reduce ANSP. Third, with regard to the method use, the practical interpretation of spatial factors has been emphasized with application scalability. Our study complements the empirical spatial evidence of ER and digitization across the change of spatial position."
In terms of "Model Selection", the content is revised as follows. For instance, "The geographic detector method is a statistical method based on the heterogeneity of a particular attribute among different geographical fields to explore the driving effect of various explanatory factors on the factors to be explained. Its advantages are numerous. First, the model has no linear hypothesis, which can better overcome the multicollinearity problem and endogeneity problem. Second, we should pay attention to the objective fact that the "sample" is located in the geographical space, emphasize the spatial heterogeneity of the intra-layer variance less than the inter-layer variance, and reveal the driving factors behind it based on this spatial heterogeneity. Third, the impact of two explanatory factors' interaction on the explained factor is detected. The identification method of exchange in the regression model is multiplicative, the interaction of two elements is not necessarily multiplicative. The geographical detector can calculate and compare the q value of every single factor as well as the q value after the superposition of the two elements, determining whether there is an interaction between the two factors and the strength and direction of the interaction [38, 39]. Therefore, this study primarily uses the factor detection. Factor interaction detection is used to test the driving effects of ER, digitalization, and their interaction on the prevention of ANSP."(in Line 388-404)
#3 In Section â…¢, there is no description to the type and processing methods of the variables. It is suggested to supplement the processing methods of different types of variables (eg. numerical quantity, type quantity).In addition, agricultural non-point source pollution has spatial heterogeneity, and the spatial distribution map of factors are not seen in the paper. Developer's literature should be given in “model selection”. For example, “Jinfeng WANG, Chengdong XU. Geodetector: Principle and prospective[J]. Acta Geographica Sinica, 2017,72(1): 116-134.”.
Reply: Thank you for your reminder. The spatial distribution map of agricultural non-point source pollution in 2010 and 2020 has been made, such as figure 1. These references have been added.
#4. The full English name notes should be given for the first abbreviation. Specifically, P3L101- 102 “ICT” and P4L194 “EP”.
Reply: OK! "EP" refers to public participation in environmental protection. "ICT" refers to " Information and Communications Technology".
#5 There are some points for attention as follows:
P3L130, L132 and P4L192-193. The second “non” is missing in “non-exclusive and -competitive”.
P7L343. What is the meaning of 0.25 in the formula (1)?
P7L345. There is a space which should be removed between D and E.
P7L349 and L356.The “factor” needs to be plural.
P8L368 and P9L418. I think that “Government Protection and Governance Investment” in the first column of Table 1 and “Government Investment in Environmental Protection” of Table 3 should be the same variable.
P8L368 Table 3and P11L459 Table 6. The initial letters of variables in “Factor” are confused. It is recommended to unify.
P8L386. The “explained” of “the explained factor X” is wrong, and it should be “explanatory”.
P9L393. “S or Sh” doesn’t appear in the formula (2), it should be “s or sh”.
P10L429 and P11L459. What regions are the "East, Middle and West" in Tables 4 and 6, respectively? Are they China's three major economic belts? It is suggested to give notes.
P12L490-491. No note is given for “OF” in Table 7.
Reply: Thank you for your reminder. These points have been revised correctly.

Reviewer 2 Report
This paper analyzes the effect of environmental regulation on agricultural non-point source pollution in rural areas of China from 2010 to 2020 by using geographical detector tool. The indicator of environmental control is too simple. This paper is innovative to some extent, but the result analysis is slightly insufficient and lacks relevant discussion.
Author Response
This paper analyzes the effect of environmental regulation on agricultural non-point source pollution in rural areas of China from 2010 to 2020 by using geographical detector tool. The indicator of environmental control is too simple. This paper is innovative to some extent, but the result analysis is slightly insufficient and lacks relevant discussion.
Reply: Thank you for your reminder. The indicators of environmental control refer to three factors such as government investment in environmental protection, government attention, and rules and regulation, which measured basis has been verified (in reference [5] and [36]).To ensure the sufficient of the result analysis, the spatial distributions of environmental regulation and digitization have been added such in figure 2 and figure 7 with something relevant discussions.

Reviewer 3 Report
This is a valid research examining how government environmental regulation and digitalization are associated with the ANSP, using province-based panel data. Area of this research matters for environmental regulation enhancement and following suggestions are for the authors to consider:
This research did lay a foundation to understand how existing or previous regulation affects agricultural non-point source pollution reduction, by some selected variables at national level, yet it is still insufficient to answer the question of “how to reduce ANSP”, may the author reconsider the title for this research given the research content.
The review is logically presented and gives hints for hypotheses, but the conclusion echos little to the review, e.g. given the quantified result, do the concerns and mechanisms the review mentioned actually happen? Maybe some explanation on how the selected variables relate to the ANSP reduction can be helpful.
Also, one can always draw some recommendations for regulation improvement, the key point is, how can findings from this research generate some unrecoganized one, e.g. can the differed impacts on east, middle and west imply differed regulation intensity toward these area? Probably a study is unnecessarily to be an all inclusive one, summarizing something different from the work may be more helpful than simply propose some recommendations.
Author Response
#1 This research did lay a foundation to understand how existing or previous regulation affects agricultural non-point source pollution reduction, by some selected variables at national level, yet it is still insufficient to answer the question of “how to reduce ANSP”, may the author reconsider the title for this research given the research content.
Reply: Thank you for your reminder. The title has been changed to " Preventing Agricultural Non-point Source Pollution in China:An Effect of Environmental Regulation with Digitization". Also, we supplement the spatial distribution (figure 2-7) of environmental regulation and digitization to answer the question of "preventing ANSP". Compared with the trend of environmental regulation or digitization and agricultural non-point source pollution, the prevention effect of ANSP can be identified with the consideration for the geographical detectors.
#2 The review is logically presented and gives hints for hypotheses, but the conclusion echos little to the review, e.g. given the quantified result, do the concerns and mechanisms the review mentioned actually happen? Maybe some explanation on how the selected variables relate to the ANSP reduction can be helpful.
Reply: Thank you for your reminder. To ensure the quantified result the reliability, the spatial distributions of environmental regulation, digitalization, and ANSP have been presented in figure 1-7. The emergence of the mentioned review is an event with the geostatistical probability. The explanation on how the selected variable related to the ANSP can be viewed in Section 3.2.1 and 3.2.2.
#3 Also, one can always draw some recommendations for regulation improvement, the key point is, how can findings from this research generate some unrecoganized one, e.g. can the differed impacts on east, middle and west imply differed regulation intensity toward these area? Probably a study is unnecessarily to be an all inclusive one, summarizing something different from the work may be more helpful than simply propose some recommendations.
Reply: Thank you for your reminder. We also added:
“According to the interpretation of the National Development and Reform Com-mission, the division of the east, the middle and the west of China is a policy division, not an administrative division, nor a geographical definition. Therefore, the east refers to the provinces and cities that have implemented the coastal opening policy earlier and have a high level of economic development, including Beijing, Tianjin, Hebei, Liaoning, Shanghai, Jiangsu, Zhejiang, Fujian, Shandong, Guangdong and Hainan; The central part refers to the economically underdeveloped areas, including Shanxi, Inner Mongolia, Jilin, Heilongjiang, Anhui, Jiangxi, Henan, Hubei and Hunan; The west re-fers to the underdeveloped western regions, including Sichuan, Guizhou, Yunnan, Ti-bet, Shaanxi, Gansu, Qinghai, Ningxia, Xinjiang and Guangxi”.
We have strengthened the theoretical interpretation of the empirical analysis part to explain the theoretical connotation of the data and strengthen the connection between theory and empirical analysis. In addition, we have added relevant suggestions in view of the regional differences in the empirical conclusions.
For instance, “Fourth, for the eastern and central regions, we should focus on strengthening the central government's cross-sectoral and cross-local coordination role in regional environmental policies, and make full use of guidance, regional planning, policy documents and other ways to strengthen sectoral and regional cooperation, and promote cross-regional comprehensive governance. At the same time, we should strengthen the publicity of digital inclusive finance in rural areas, effectively improve farmers' understanding and acceptance of digital inclusive finance, improve financial availability, and ensure that digital inclusive finance and green finance play a better role in environmental protection. For the western region, it is the premise to strengthen the support of relevant departments for the construction of rural digital infrastructure in the western remote areas, strengthen the publicity and demonstration of digital rural application scenarios, and enhance the awareness and willingness of rural residents to digital technology.

Round 2
Reviewer 1 Report
The authors have addressed my concerns, and the manuscript is much better now.
Author Response
This manuscript has been updated to perfect the content.
The graphs to map each factor's spatial distribution have been grouped into two figures consisting of figure 2 representing ER and figure 3 reflecting diginization.
Also, the words like "we should focus...." and " we should support......." have been revised. For instance, it is noted that "the central planners' cross-sectoral and cross-local coordination role in regional environmental policies utilize guidance, regional planning, policy documents, and other ways to strengthen sectoral and regional cooperation could be improved to promote cross-regional comprehensive governance" and "the policy that supports the publicity of inclusive digital finance in rural areas, effectively improves farmers' understanding and acceptance of inclusive digital finance, as well as financial availability, and ensuring that inclusive digital finance and green finance play a better role in environmental protection".
Additionally, the word expression of policy recommendation has been changed, such as "is suggested to be given", "The implementations consisting of", "could be", "is taken to", and "the policymakers could", as shown in red words of Section 5.2.
